# Associations of Neonatal Dairy Calf Faecal Microbiota with Inflammatory Markers and Future Performance

**DOI:** 10.3390/ani14172533

**Published:** 2024-08-31

**Authors:** Marina Loch, Elisabeth Dorbek-Sundström, Aleksi Husso, Tiina Pessa-Morikawa, Tarmo Niine, Tanel Kaart, Kerli Mõtus, Mikael Niku, Toomas Orro

**Affiliations:** 1Institute of Veterinary Medicine and Animal Sciences, Estonian University of Life Sciences, F. R. Kreutzwaldi 62, 51006 Tartu, Estonia; elisabeth.dorbek-sundstrom@emu.ee (E.D.-S.); tanel.kaart@emu.ee (T.K.); kerli.motus@emu.ee (K.M.); toomas.orro@emu.ee (T.O.); 2Veterinary Biosciences, Faculty of Veterinary Medicine, University of Helsinki, Agnes Sjöbergin katu 2, P.O. Box 66 Helsinki, Finland; aleksi.husso@helsinki.fi (A.H.); tiina.pessa-morikawa@helsinki.fi (T.P.-M.); mikael.niku@helsinki.fi (M.N.)

**Keywords:** neonatal dairy calf, average daily weight gain, first lactation dairy cow performance, calf faecal microbiota, 16S rRNA gene amplicon sequencing, acute phase protein

## Abstract

**Simple Summary:**

Newborn animals come into contact with bacteria and other microbiota for the first time, and their immune system has to adapt to their presence. The resulting immune response and possible symbioses with certain bacterial genera affect the animal’s future performance. In this study, specific bacterial genera were detected in the faeces of newborn dairy calves under four weeks old whose abundance was associated with inflammatory marker concentrations during this period, as well as growth, reproductive health, and fertility up to three years later. The findings indicate that microbiota can influence the inflammatory response and through this, possibly the future performance of the dairy heifer.

**Abstract:**

After birth, the immune system is challenged by numerous elements of the extrauterine environment, reflected in fluctuations of inflammatory markers. The concentrations of these markers in the first month of life are associated with the future performance of dairy youngstock. It is thought that bacterial genera colonizing the calf intestinal tract can cause inflammation and thus affect their host’s performance via immunomodulation. This study explored how the faecal microbiota of newborn dairy calves were related to inflammatory markers during the first three weeks of life, and if the abundance of specific genera was associated with first-lactation performance. Ninety-five female Holstein calves were studied. Once a week, serum and faecal samples were collected, serum concentrations of serum amyloid A, haptoglobin, tumour necrosis factor-α, and interleukin-6 were measured, and faecal microbiota composition was examined by 16S rRNA gene amplicon sequencing. Faecal *Gallibacterium* abundance in the first week of age and *Collinsella* abundance in the second week were negatively associated with inflammatory response as well as with calving–conception interval. *Peptostreptococcus* abundance in the second week of life was positively associated with inflammatory response and calving–conception interval, and negatively with average daily weight gain. In the third week, *Dorea* abundance was positively, *Bilophila* abundance was negatively associated with inflammatory response, and both genera were negatively associated with age at first calving. These bacterial genera may be able to influence the inflammatory response and through this, possibly the future performance of the dairy heifer. Deciphering such microbiota–host interactions can help improve calf management to benefit production and welfare.

## 1. Introduction

Tissue damage and invasion by pathogens cause an inflammatory response, which includes the release of pro-inflammatory cytokines such as interleukins (IL), tumour necrosis factor-alpha (TNF-α), and interferons. These cytokines trigger the synthesis of acute phase proteins (APPs) in the liver and sometimes in local tissues [1]. This process is called the acute phase response (APR), and as the serum concentrations of cytokines and APPs increase significantly in response to a threat, they can be used as unspecific inflammatory markers [2], as well as markers of stress in calves and sheep [3,4]. In cattle, serum amyloid A (SAA) and haptoglobin (Hp) are the APPs that increase the most during APR [5,6].

In addition, it has been found that neonatal ruminants undergo periodical fluctuations in these serum concentrations not related to infections and without showing clinical symptoms [7,8,9,10]. Generally, IL-6 and TNF-α serum concentrations in dairy calves are highest during the first week and decrease afterwards. Hp serum concentration is already high at birth, and shows a peak during the second week, before declining. SAA serum concentration is low on the day of birth, elevated during the first two weeks, and declines from the third week of life [10,11]. 

Although it can be hypothesized that the fluctuation of APPs and cytokines in the neonatal period reflects a physiological adaptation process(“immune priming”), as liver function develops and the immune system matures, there could be harmful consequences for the animal when the concentrations of these inflammatory markers are particularly high during the first weeks of life. Higher concentrations of SAA and Hp during the second (and to a lesser degree in the third) week of life are associated with lower average daily weight gain (ADWG), higher age at first calving (AFC), and longer calving–conception interval in heifers [12]. Higher concentrations of TNF-α and IL-6 during the first week of life are associated with an increased risk of reproductive issues such as endometritis during the first lactation, and IL-6 concentration in the first and second weeks of life is positively associated with calving–conception interval [12]. This indicates that the APPs and cytokines could be markers for underlying processes affecting long-term performance. Productive performance and good overall health during the first lactation in turn are indicators of the performance in future lactations and the productive life-length of dairy cows [13,14,15].

Since the increase in concentrations of inflammatory markers in the first weeks of life is not necessarily related to clinical infection, the question of its cause arises. The birth process itself is a stressful event, however, the difficulty of birth did not cause differences in APP concentrations [10]. Colostrum contains all four of the aforementioned markers, but only TNF-α and IL-6 concentrations are positively associated in colostrum and calf serum beyond the first week of life [10]. Hp concentration in colostrum does not affect serum Hp concentration in offspring [10,16,17]. SAA concentration in colostrum was negatively associated with SAA concentration in calf serum during the first week of life, but not later [10]. Thus, colostrum APP concentrations can influence the offspring’s APR to some extent, but they do not seem to be the main factor.

Another important aspect of adaptation to life outside the uterus is colonization of the host with microbiota. The most intensive colonization occurs during the first week of life, due to free niches [18]. At the same time, the colostrum influence on APR is still stronger than later on, providing direct protection as well as affecting the calves’ cytokine and APP concentrations [10]. In summary, during the second week of life, the concentrations of SAA and Hp peak, and colostrum influence on APR decreases. In the third week, concentrations of cytokines and APPs in the serum of calves begin to decrease, potentially indicating that the immune system has started to adapt to its challenges and the microbiota to their host environment. 

The healthy uterus contains few, if any microorganisms, so mammals encounter a much bigger and new array of microorganisms during and after birth [19,20,21,22,23]. The relationships between intestinal microbiota and host development and immune response are the subject of a growing field of research. In humans and mice, these relationships have been studied more thoroughly than in production animals. Some of the uncovered mechanisms may be similar in calves, as they function similarly to monogastric mammals while on a milk-only diet [24].

For example, high abundances of the anaerobic genera *Bifidobacterium* and *Lactobacillus* are found in the calf intestine, and their interactions with the mucosa may promote the development of tight junctions, as the epithelial permeability decreases soon after birth [25,26,27]. These two bacterial genera may also play a role in the control of the inflammatory response to other microbiota, as their presence stimulates the synthesis of interleukin-10 (IL-10), which inhibits the production of pro-inflammatory cytokines [27,28]. More specifically, *Bifidobacterium infantis* was shown to normalize epithelial permeability and increase transepithelial resistance in a mouse model with colitis [29]. Germ-free animals showed a much thinner colon wall and reduced expression of toll-like receptors (TLRs) in the small intestine compared to normal controls [30,31]. *Lactobacillus plantarum* was found to stimulate TLR-2 and increase scaffold proteins in the vicinity of tight junctions as well as to help maintain optimal somatic growth [32,33]. In three-week-old calves, the expression of TLR-2 in the jejunum and TLR-6 in the ileum and cecum were negatively correlated with the total bacterial population in ingesta; and TLR-2, 6, and 9 were negatively correlated with mucosa-associated bacterial count [34]. It has been suggested that commensal bacteria can regulate TLR activity, so that inflammatory responses are initiated towards pathogens only [35].

In summary, colonization of the intestines with microbiota plays an important role in the maturation of the immune system and thus subsequent resistance to disease, as well as growth. The colonization may also cause APR without causing clinical symptoms, as the host arranges its symbiosis with commensals. SAA for example has been shown to increase upon microbiota colonization of germ-free zebrafish, leading to increased neutrophil recruitment to extra-intestinal injuries but also suppressing bactericidal activity of neutrophils [36,37]. SAA gene expression is also significantly downregulated in germ-free foetal mouse intestines compared to normal controls [38].

As the concentrations of inflammatory markers in the neonatal period are associated with future performance [10,12,39], it can be hypothesized that certain bacterial genera found in the faecal microbiota of neonatal calves may influence future performance and health through immunomodulation, specifically, via their relationship with APR.

The aims of this study were to describe the faecal microbiota of neonatal calves during the first three weeks of life, to investigate associations of microbiota abundances with APR, and to determine if genera showing such associations may have associations with production performance and health during the first lactation.

## 2. Materials and Methods

This observational study is part of a large-scale study using the same raw data [10,12,39,40]. 

### 2.1. Animals

This cohort study was conducted on a large commercial dairy farm in Central Estonia, and all female calves born from 21 January to 16 March 2015 were included (*n* = 144). At the time, approximately 1900 milking cows lived on the farm, each producing an average of 10,000 kg of milk per year [41]. The details of housing, feeding, and management of the animals have been described previously [10,39,40]. In short, the calves were removed from their dams immediately after birth, weighed on a digital scale, and fed 3 L of quality-controlled colostrum from their own dams within two hours after birth. The quality was assessed with a colostrum densimeter (Jørgen Kruuse A/S, Langeskov, Denmark) by the farm employees and deemed sufficient if the specific gravity was >1.035. In one case, quality was insufficient, so frozen colostrum from another cow on the same farm was used.

During the first four weeks of their lives, the calves were housed individually, then moved to group pens of 8–10 calves with sawdust bedding on the concrete floor. In addition to ad libitum access to hay and prestarter feed (Prestarter, Agrovarustus OÜ, Tartu, Estonia), and later starter feed from the same manufacturer the calves received 2–3 L of warm, unpasteurized raw milk twice per day during the first two weeks of life, later milk replacer (Josera GmbH & Co. KG, Kleinheubach, Germany; 1 L of warm water mixed with 140 g of milk powder, 3 L twice daily), which was decreased from 4 weeks of age until weaning at two and a half months. After weaning, the calves had continued access to starter feed and hay. 

All diagnoses of diseases and treatments were performed by veterinarians who were full-time employees on the farm and not involved in the sampling.

The calves were vaccinated against parainfluenza virus type three (PI3V) and bovine syncytial virus (Rispoval RS+PI3 Intranasal, Zoetis Belgium SA, Louvain-la-Neuve, Belgium) at two days of age, and against bovine herpesvirus-1 (Hiprabovis IBR Marker live, Laboratorios HIPRA SA, Girona, Spain) at three months of age. At one to two months of age, the calves received a single prophylactic treatment against *Eimeria* spp. (Cevazuril, Ceva Santé Animale, Libourne, France).

The herd was positive for bovine viral diarrhoea virus antibodies and had previously tested positive for coronavirus, rotavirus, and *Escherichia coli* F5 antigen (personal communication with farm veterinarians). After the sampling for the study had started, the farm veterinarians observed increased mortality associated with diarrhoea and determined cryptosporidiosis as the cause. Mass treatment with halofuginone lactate (HL; Halocur, Intervet International B. V., Boxmeer, The Netherlands) was initiated for all calves under two weeks old. According to the manufacturer, calves should receive one dose of HL daily for the first seven days of life, starting no later than 48 h after birth for the prophylactic treatment protocol. Some animals included in the study were already older than two weeks and thus received no treatment, others received treatment despite being older than 48 h, thus classified as “incorrect treatment”, and calves born after the diagnosis of cryptosporidiosis in the herd (and the initiation of the mass treatment with HL) were treated according to the abovementioned protocol (classified as “correct treatment”). This was later considered in the statistical analysis, as was *Cryptosporidium* spp. oocyst shedding. As *Cryptosporidium* spp. was identified as a cause of the acute diarrhoea, other possible agents were not tested for. 

The animals were weighed again on a digital scale at approximately 12 months of age. The first artificial insemination was performed when the cows had reached a body weight of 370 kg and were at least 365 days old. All milk and health data were recorded on the farm using DairyComp (VAS, Tulare, CA, USA). The number of noted health problems was acquired from the farm records, of which reproductive issues (abortion, endometritis, retained placenta) were later analysed.

Not all animals originally included in the study could be analysed for each outcome. Twelve calves died during the first three weeks of life [related to diarrhoea (*n* = 9), respiratory symptoms (*n* = 2) and navel infection (*n* = 1)]. Before weaning, a total of 13.8% of the animals died. After the heifers had reached one year of age, some of them were sold for slaughter due to issues related to fertility and lameness. The number of animals decreased for each outcome due to different survival times and varied due to sample availability (Table 1).

### 2.2. Data and Sample Collection

Faecal and blood samples were collected once per week from all heifer calves born during the study period, but not all calves were available for sampling each time. Consequently, three age groups of heifer calves were studied separately: W1 (first week of life, covering an age span of 1 to 7 days of age), W2 (second week of life, 8 to 14 days of age), and W3 (third week of life, 15 to 21 days of age). Blood samples were collected into sterile evacuated test tubes (VACUETTE^®^ TUBE 9 mL CAT Serum Clot Activator) with an 18-G sterile needle from the jugular vein. Samples were centrifuged (1800× *g*, 10 min) and the serum was separated and stored in aliquots at −20 °C until further analysis. Faecal samples were collected directly from the rectum using disposable gloves and kept at 4 °C for up to 48 h until flotation for detection of *Cryptosporidium* spp. oocysts and then at −20 °C until DNA extraction. If calves did not have faecal matter available at the time of blood sampling, these animals were not caught again to avoid excessive stress due to catching and restraining more than once a day, and the serum samples were excluded from statistical analysis. Furthermore, some faecal samples were too low in volume to extract DNA after parasitological analysis, which was especially the case in W1. Thus, the number of available samples varied by week (Table 1, Appendix A). 

### 2.3. Laboratory Analysis

Serum SAA and cytokine concentrations were measured with commercial ELISA kits (bovine SAA Phase BE kit—Tridelta Development Ltd., Dublin, Ireland; bovine IL-6 and TNFα—Cusabio Biotech, Wuhan, Hubei, China) according to the manufacturers’ instructions. For Hp measurements, the colorimetric method developed by Makimura and Suzuki [42] was modified following Alsemgeest et al. [43], using microtitration plates, and tetramethylbenzine (60.0 mg/L) as the substrate. The minimal detection limits were the following: 0.3 mg/L for SAA, 60.0 mg/L for Hp, 2.5 ng/L for IL-6 and 50.0 ng/L for TNF-α. The inter- and intra-assay coefficients of variability for all protein detection methods were <15%.

*Cryptosporidium* spp. oocyst count was determined using an immunofluorescence method described by Niine et al. [40]. Results were expressed as an approximate number of oocysts per gram of faecal matter (opg).

DNA from faeces was extracted using the PSP^®^ Spin Stool DNA Kit (STRATEC Biomedical AG, Birkenfeld, Germany) according to the manufacturer’s instructions. The V3-V4 region of the 16S rRNA gene was pre-amplified with 12 PCR cycles, and then sequenced using the Illumina MiSeq platform in the DNA core facility of the University of Helsinki. The raw sequencing data were processed using the ASV-based QIIME2 pipeline, as described previously [26,39,44]. A detailed description of the microbiota analysis can be found in the Supplementary Material of Dorbek-Kolin et al. [39]. 

### 2.4. Statistical Analysis

All statistical analyses were performed separately by calves’ age groups, i.e., week of life, as different effects and mechanisms of the associations were hypothesized for each week of age. One sample per animal per week was available, but the exact day of age within the week varied. Although all days of age were covered in each week, the different number of daily samples, the individuality of animals and relatively small sample size did not allow us to perform longitudinal modelling by days. Thus, analysis by week was deemed more robust and the exact age in days was only included in models as a confounder to reduce the within-week variability.

For statistical analysis, the calves were retrospectively divided into groups based on their infection status and treatment with HL (Appendix A): “no treatment”, “incorrect treatment”, and “correct treatment”; as well as no *Cryptosporidium* spp. oocysts found low oocyst count (below median opg of the week group in question), and high oocyst count (above median opg of the week group in question).

To reduce data dimensionality, genera were selected for analysis in multiple steps as described below, an overview of which is provided in Figure 1.

In the present study, the “core microbiota” were defined as genera with a relative abundance of ≥0.1% in ≥10% of samples of the respective week group. In W1, the core microbiota consisted of 39 genera, in W2 of 62, and in W3 of 84 genera. 

To distinguish between a strong and a weak inflammatory response, three evenly-sized groups were established for each inflammatory marker based on the data for each week group: low, moderate, and high concentrations (Table 2). This was a data-driven approach, as no biological threshold values are established, and each week can be assumed to be different due to the age-related fluctuations in concentrations and the decreasing influence of colostrum [10]. Random forest analysis was used to show the most influential bacterial genera on the APR. The top third influential core genera (W1: number of genera (n_g_) = 13, W2: n_g_ = 20, W3: n_g_ = 28) for each marker according to the random forest analysis were then further analysed using negative binomial regression. In the regression models, genus abundance was the outcome, inflammatory marker concentration group was the explanatory variable, and *Cryptosporidium* spp. oocyst count group, HL treatment group, and age in days (categorical) were added as potential confounders. Total 16S sequencing read counts in the sample were included as an offset variable. Wald test *p*-values were calculated for the inflammatory marker groups for each model, and Holm–Šidak adjusted *p*-values were calculated to account for multiple comparisons. For genera with significant Holm–Šidak corrected Wald test (α = 0.05), backward stepwise elimination procedure was performed to obtain the final model with a significant association between genus abundance and inflammatory marker concentration groups. Variables were considered confounders if their removal changed the coefficient of the main explanatory variable by more than 10%. Bonferroni-corrected *p*-values were used for pairwise comparisons between three-level grouping variables.

In the second phase of analysis, relationships between microbiota and performance were explored. Here, genera whose abundances were significantly associated with inflammatory marker concentration were included. 

Based on the outcome variables’ distribution, linear regression (ADWG, AFC, 305-day milk yield), logistic regression (occurrence of reproductive diseases), and negative binomial regression (calving–conception interval) models were built. The explanatory variable was logarithmically transformed genus abundance. Alpha diversity, measured in the Shannon diversity index, was also investigated as an explanatory variable for each outcome, but was not associated with inflammatory marker concentrations nor performance outcomes.

As in the first phase, each week group was again analysed separately, and within each week, all genera that had shown significant associations with an inflammatory marker concentration in that week group were included in the same model. Exact age in days (to account for age effect within week group), HL treatment group, *Cryptosporidium* spp. oocyst count, and parity of the dam were included as possible confounders. For ADWG, birth weight was used instead of dam’s parity; for milk yield and calving–conception interval, AFC was included as an additional confounder. The final models were obtained using stepwise backward elimination of insignificant (*p* > 0.05) independent variables if their removal did not change the coefficient of the explanatory variable by more than 10%.

Basic data management was performed using Excel 2016 (Microsoft, Redmond, WA, USA) and Python 3.5.1 (Anaconda 4.0.0, Continuum Analytics, Austin, TX, USA). The package ‘randomForest’ [46] with R version 4.0.1 (R Foundation for Statistical Computing, Vienna, Austria) was used. Regression analysis was performed in STATA/IC 14.2 (StataCorp LP, College Station, TX, USA). The results were interpreted as significant if *p* ≤ 0.05.

## 3. Results

The microbial alpha diversity, measured as Shannon diversity index, increased with each week of age: W1 (mean ± SD: 3.2 ± 1.1), W2 (3.9 ± 1.2), W3 (5.2 ± 0.9) (Figure 2). 

The core microbiota of W1 (number of samples (n_s_) = 67) consisted of 39 genera, belonging to the phyla *Pseudomonadota* (relative abundance 34.1%), *Bacteroidota* (23.8%), *Firmicutes* (23.2%), *Actinomycetota* (7.2%), and *Fusobacteriota* (11.7%). In W2 (n_s_ = 95), the core microbiota consisted of 62 genera, belonging to the phyla *Firmicutes* (51.0%), *Bacteroidota* (29.5%), *Fusobacteriota* (8.8%), *Pseudomonadota* (6.6%), *Actinomycetota* (3.9%), and *Verrucomicrobiota* (0.2%). The core microbiota of W3 (n_s_ = 83) consisted of 84 genera, belonging to the phyla *Firmicutes* (46.6%), *Bacteroidota* (37.8%), *Actinomycetota* (6.0%), *Fusobacteriota* (4.7%), *Pseudomonadota* (4.1%), and *Desulfobacterota* (0.7%) (Figure 3). 

All following associations are examined at the genus level.

### 3.1. First Week of Life

For complete model results, see Appendix A.

In W1, an abundance of *Erysipelatoclostridium* was positively associated with the serum SAA concentration group. Abundances of *Escherichia-Shigella* and *Megasphaera* were positively associated with the IL-6 concentration group (Figure 4A), and the abundance of *Megasphaera* was positively associated with calving–conception interval (*p* = 0.028; Figure 5A). The abundance of *Gallibacterium* was negatively associated with the IL-6 concentration group (Figure 4B) and calving–conception interval (*p* = 0.014; Figure 5B).

### 3.2. Second Week of Life

For complete model results, see Appendix A.

The abundance of *Succinivibrio* was significantly higher in the moderate SAA group than in the low and high concentrations groups (*p* = 0.009 between low and moderate, *p* = 0.003 between moderate and high). The abundance of *Collinsella* was negatively associated with SAA and Hp concentration groups in W2 (Figure 4C,D) and with calving–conception interval (*p* < 0.001; Figure 5C). The abundance of *Flavonifractor* was negatively associated with the Hp concentration group. The abundance of the genus [*Eubacterium*] *coprostanoligenes* group was negatively associated with the IL-6 concentration group. Abundance of *Peptostreptococcus* was positively associated with the Hp concentration group (Figure 4E) and calving–conception interval (*p* = 0.010; Figure 5E), and negatively with ADWG (*p* = 0.002; Figure 5D). Abundance of the genus *Streptococcus* was significantly higher in the moderate than in the low Hp group (*p* < 0.001). 

### 3.3. Third Week of Life

For complete model results, see Appendix A.

In W3, an abundance of *Dorea* was positively associated with the Hp concentration group (Figure 4F). *Bilophila* abundance was negatively associated with the TNF-α concentration group (Figure 4G). Abundances of *Bilophila* and *Dorea* were negatively associated with AFC (*p* = 0.050; *p* = 0.021, respectively; Figure 5F,G). The abundance of *Erysipelotrichaceae* UCG-004 was negatively associated with IL-6 and TNF-α concentration groups.

## 4. Discussion

In this study, the faecal microbiota of newborn dairy calves on the genus level was investigated in their relationship with systemic inflammatory markers and future performance. 

During the first three weeks of the calves’ lives, faecal microbiota alpha diversity increased. This reflects the colonisation of the gastrointestinal tract taking place in the first weeks of life. Previous studies also reported an increase in alpha diversity with age [26,47,48].

The majority of genera in each week group belonged to the phyla *Firmicutes*, *Bacteroidota*, and *Pseudomonadota*. This is in accordance with previous studies [26,47,49]. 

Microbiota–host interactions in early life play an important role in immune development and thus can affect future immune responses [35,50]. The relationship of bacterial genera abundances with inflammatory response changes during the first three weeks of life, and so do associations of inflammatory marker concentrations with future performance [12]. Paralleling these changing relationships of inflammatory marker concentrations with future performance by week of life, in each week group in the present study, different genera had associations with future performance, and this may be due to an immunomodulating effect of the bacteria, specifically on the APR, that impacts the hosts’ performance. 

Although regarded as a marker of innate immunity, SAA also induces adaptive immune responses, as it attracts type 2 T helper (Th2) cells and acts as a soluble pattern recognition receptor for these cells [51,52]. Thus, SAA serves as a marker for Th2 activation, which may affect future immune responses and metabolism. Newborns generally show a bias towards Th2 response (with low type 1 T helper (Th1) cell activity), which can be enhanced by SAA [53]. Both SAA and IL-6 have been linked to an increase in Th2 cells at the expense of Th1 [54,55]. As bacterial colonization [37], and, in the present study, specifically a high abundance of the genera *Erysipelatoclostridium*, *Escherichia-Shigella*, and *Megasphaera* are associated with increased concentrations of SAA and IL-6, these bacteria may be able to shift the immune response to a Th2 dominance. Th2 response is characterized by systemic inflammation, which can cause reduced weight gain due to lack of appetite [56]. If the microbiota stabilizes at a composition favourable to a Th2 bias, the bias could also persist and lead to a persistently reduced appetite [56], which could lead to decreased weight gain compared to animals with a microbiota related to lower IL-6 and SAA and more Th1. *Flavonifractor*, on the other hand, has been found to suppress the Th2 response [57], which could mean that a high abundance of *Flavonifractor* might lead to increased weight gain. Hp concentrations in the second week of life have been negatively associated with nine-month-ADWG [10], meaning that *Flavonifractor*, in this study negatively associated with Hp concentrations in W2, might contribute to weight gain. 

The abundance of *Peptostreptococcus* in W2 was negatively associated with ADWG, and with increased inflammatory markers in the same week (current study and [39]), meaning that its interactions with the immune system could lead to persistently lower weight gain. *Peptostreptococcus* has been found to be involved in the regulation of body weight. Its abundance is higher in obese humans than in those with normal weight [58], but the current study found a higher abundance in calves with lower ADWG, suggesting that there is an optimal abundance of *Peptostreptococcus* (and other genera) for maintaining a healthy weight. Furthermore, in a recent study from the USA, *Peptostreptococcus* was one of two genera that characterized the faecal microbiota of diarrhoeic 2–3 week-old Holstein calves in contrast to age-matched healthy calves, whose microbiota was in turn characterized by *Collinsella aerofaciens* [48]. Similarly, in 21-day-old calves, *Peptostreptococcus* was among the genera with a higher relative abundance in diarrheic calves compared to clinically healthy age-matched calves, who had a higher relative abundance of *Collinsella* [59]. A high abundance of the genus *Collinsella* seems to be beneficial to dairy heifers. In the present study, *Collinsella* abundance was negatively associated with SAA and Hp concentrations as well as with calving–conception interval in W2. A study in humans supports the assumption that a high abundance of *Collinsella* early in life is related to weight gain over an extended period of time [60]. Growing well and maintaining a healthy weight are major factors for fertility in cows [61,62], indicated by a shorter calving–conception interval compared to animals that had a lower abundance of *Collinsella* in the neonatal period. In a study of two-week-old calves by Castro et al. [63], greater relative abundances of *Streptococcus*, *Faecalibacterium*, *Oscillospira*, *Collinsella*, and *Clostridium* were related to higher concentrations of short-chain fatty acids in the intestines and better growth performance until eight weeks of age compared to calves with higher abundances of *Lactobacillus* and *Bifidobacterium*. The differences in microbiota composition were significant only at this time point, not at four weeks of age, and thus, the authors suggested a narrow time window for prebiotics to be effective [63]. *Collinsella* produce ursodeoxycholic acid, which has been shown to reduce gene expression of IL-6 and TNF-α [64], and by that also the expression of APPs. Hence, *Collinsella* may be able to reduce the inflammatory response and increase weight gain and improve fertility through a mechanism that includes this metabolite, even though the current study did not find a significant association between *Collinsella* abundance and ADWG.

The genera that are regarded as beneficial for weight gain and immune homeostasis of calves in previous studies [65,66], namely *Bifidobacterium*, *Lactobacillus*, and *Faecalibacterium*, did not show any significant associations with outcomes of interest in this study. However, abundances of these genera were positively correlated with each other as well as with abundance of *Collinsella*, and negatively correlated with *Peptostreptococcus* abundance.

*Gallibacterium* abundance in W1 was also associated with a shorter calving–conception interval, and with lower IL-6 concentration in serum. Like *Collinsella*, it may affect future fertility via immunomodulation. *Gallibacterium* has been found in bovine faeces previously, but not much is known about its relationship with the bovine host [67,68]. A study by Gomez et al. [69] found *Gallibacterium* to be among the genera that were higher in abundance in two to three-week-old diarrheic calves than in healthy ones. *Gallibacterium* has been found in calves with bronchopneumonia, but is best known as an opportunistic respiratory pathogen in birds, where it reduces production and increases mortality [70]. In the present study, a significant negative association with inflammatory markers was found, but only in W1. This indicates a changing relationship between the genus and APR over time, for example as opportunists eliciting a weak inflammatory response at first, but a stronger one later, or possibly even being “anti-inflammatory” in the first week of life, but “pro-inflammatory” later.

Understanding microbiota–host relationships is necessary to develop ways to influence the health and performance of production animals for the better, for example by administering pre- or probiotics. The ideal time for such interventions is a topic of current discussions [47,71,72]. In humans, it is long known that the neonatal microbiota, meaning the first colonizers of the gastrointestinal tract, establish the conditions for subsequent microbiota as well as weight gain. For example, the abundance of *Staphylococcus aureus* and *Bifidobacterium* spp. in six-month-old infants can predict adiposity at seven years old [73,74]. 

The results of the current study show that the microbiota of dairy calves can, even in an environment with small variability between individuals, be related to future performance (including weight gain) already in the first week of life. Studies in humans have shown how disruptions in early life microbiota-immune interactions can lead to persistent immune abnormalities and increased disease susceptibility [50]. 

Limitations of this study include the incomplete records by farm veterinarians regarding diseases and their treatments of the animals regarding possible infections or other causes that may have activated the calves’ APR but were not noticed. During the first three weeks of life, umbilical or joint inflammation, respiratory disease, and other pathogens causing diarrhoea (e.g., rotavirus, coronavirus, *E. coli*), may affect the calves, and these infectious agents were not investigated and thus not considered in the statistical models. However, associations of APP concentrations in neonatal ruminants with their ADWG have been found in other studies as well, so it can be assumed that the associations between APR and future performance are due not to specific pathogens, but rather a more general immunological process [7,8,9,75,76].

There are also limitations to the generalization of this study. Firstly, the calves on this farm received a low amount of colostrum and later milk replacer, which may negatively impact their performance and might also have effects on microbiota colonization and/or inflammatory response. However, all calves received the same products and the same amounts, so this was not considered in the statistical analysis. Secondly, the samples were collected during a relatively short period of time during early spring. Thus, the results obtained may not be generalizable to a wider population and should be validated in further studies.

Cryptosporidiosis affects the microbiota composition of the calves, as has been addressed by Dorbek-Kolin et al. [39]. The parasitic infection status as well as the inconsistencies in treatment protocol were accounted for in the statistical analysis. HL is the standard treatment for cryptosporidiosis in calves, and the infection is so widespread globally, that both the presence of the protozoan as well as the HL treatment reflect the situation on many dairy farms, making the present study representative of the common environment of dairy calves [77,78].

As in any field study, unmeasured environmental factors (including during the time between sampling and outcome measures) may play a role as well. However, performing the study in only one farm has the advantage that all heifers were brought up in the same management conditions, so differences in environment or feeding were not the cause of differences in performance. All animals in the study living on the same farm during the same time frame and being of the same breed also leads to smaller variations both in microbiota composition and in performance than would be the case when investigating multiple farms or during varying seasons. Taking this into account, the relatively small effect sizes found in this study are quite considerable. In addition, a recent study showed that the differences in faecal microbiota composition of calves under 6 weeks old depend mostly on age, followed by health status, whereas differences between farms are minor [48]. Thus, similar associations can be expected on other farms, as similar microbiota compositions can be expected in calves of the same ages. As this was an observational study, no clear conclusions can be made about cause and effect, it can only be speculated that differences in microbiota abundances are behind inflammatory response and contribute to future performance differences.

While 16S rRNA gene amplicon sequencing is a commonly used method (albeit by far not the only nor necessarily the best) to identify the (faecal) microbiota of an animal, the statistical analysis of such genetic data is challenging [79]. The most pressing issue is the high dimensionality of microbiota data compared to a relatively small sample size. In this study, random forest analysis as well as the selection of a core microbiota were used to decrease the data dimensionality. Analysing a core comes with its own limitations, as for example the assignment to the core can differ vastly between studies [45], and genera that may have a big effect in a low abundance may not be analysed at all. As shown in this study, biomarkers like APPs can aid in exploring microbiota–host relationships by limiting the number of genera to be analysed in a biologically, rather than purely statistically meaningful way.

## 5. Conclusions

The observations in this study suggest that early-life faecal microbiota can influence dairy cow performance via immunomodulation. It can be assumed that specific bacterial genera (such as *Collinsella* and *Gallibacterium*) are able to downregulate inflammation and increase future performance, while others, such as *Peptostreptococcus*, can promote inflammation and decrease future performance. As the associations found in this study change over the first three weeks of life, the timing of colonization with specific bacteria seems to play a critical role.

## Figures and Tables

**Figure 1 animals-14-02533-f001:**
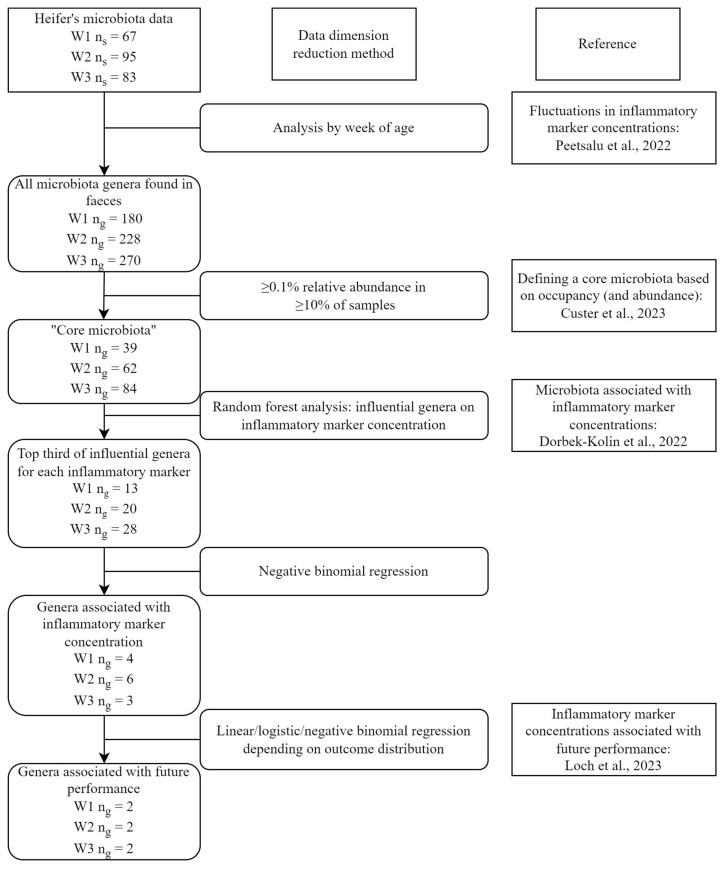
Overview of data dimension reduction and statistical analysis in this study. W1—heifer calves ages 1–7 days; W2—ages 8–14 days; W3—ages 15–21 days; n_s_—number of samples; n_g_—number of genera [10,12,39,45].

**Figure 2 animals-14-02533-f002:**
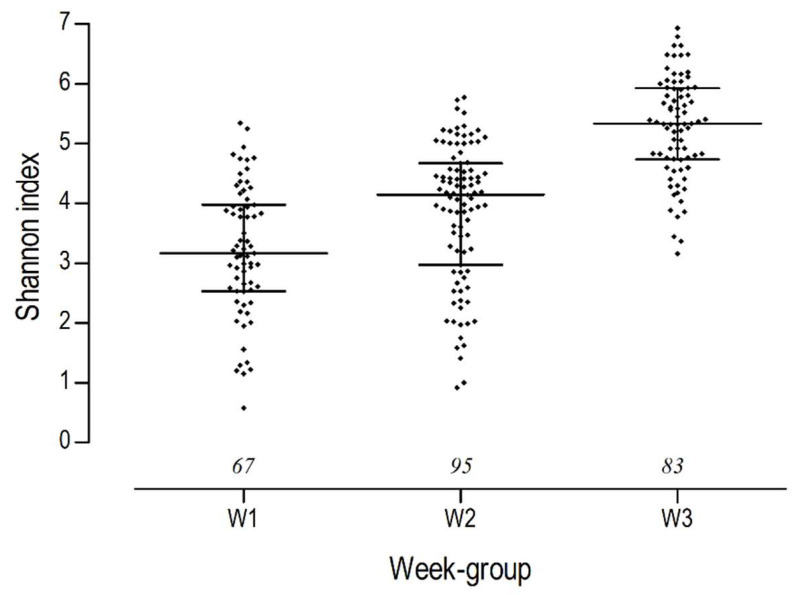
Shannon index (median and interquartile range) of faecal microbiota of newborn dairy calves during the first three weeks of life. Italic numbers above the *X*-axis represent sample size from the respective week group (W1—heifer calves ages 1–7 days. W2—ages 8–14 days. W3—ages 15–21 days).

**Figure 3 animals-14-02533-f003:**
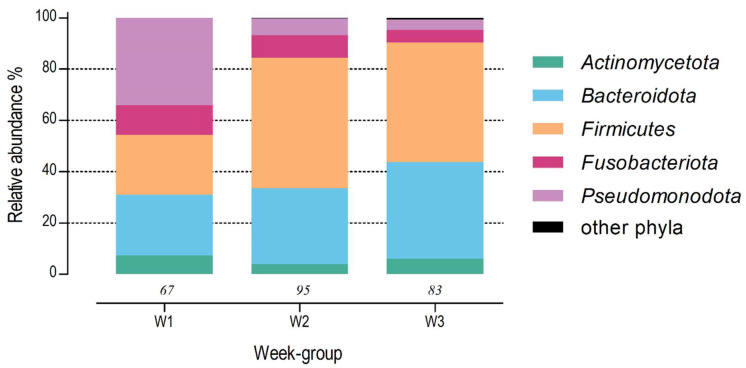
Relative abundances of “core microbiota” (relative abundance of >0.1% in ≥10% of the faecal samples in the respective week group) in faeces of newborn dairy calves at phylum level in W1 (ages 1–7 days), W2 (ages 8–14), and W3 (ages 15–21 days). Italic numbers above the *X*-axis represent sample size per week group.

**Figure 4 animals-14-02533-f004:**
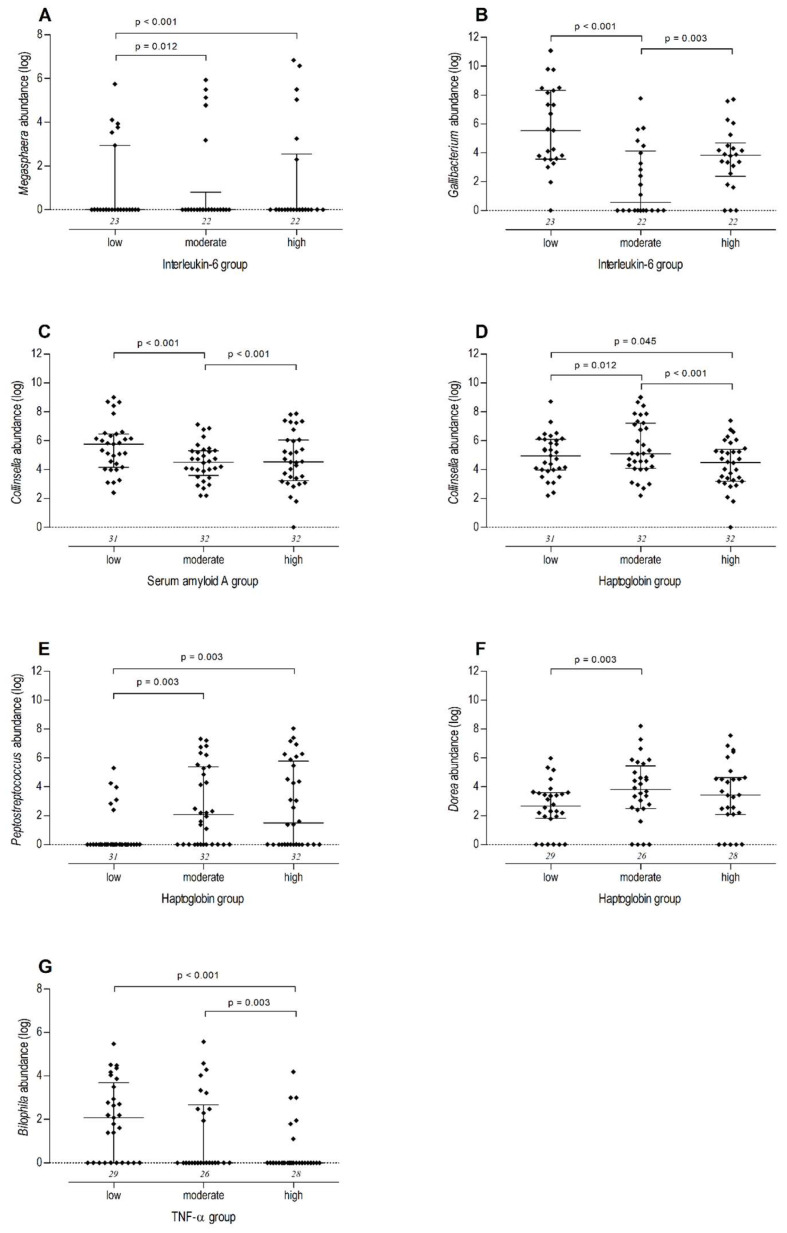
Logarithmically transformed abundances of faecal microbiota genera by inflammatory marker concentration category (median and interquartile range). Italic numbers above the *X*-axis depict sample number per concentration category. Bonferroni-corrected *p*-values are provided for statistically significant differences in genus abundance between concentration groups. Model results of multivariate regression can be found in the Appendix A. *Megasphaera* abundance in W1 (ages 1–7 days) by interleukin-6 (IL-6) concentration group ((**A**); Appendix A). *Gallibacterium* abundance in W1 by IL-6 concentration category ((**B**); Appendix A). *Collinsella* abundance in W2 (ages 8–14 days) by serum-amyloid A (SAA) ((**C**); Appendix A) and haptoglobin (Hp) ((**D**); Appendix A) concentration category. *Peptostreptococcus* abundance by Hp concentration category in W2 ((**E**); Appendix A). *Dorea* abundance by Hp concentration group in W3 (ages 15–21) ((**F**); Appendix A). *Bilophila* abundance by tumour necrosis factor alpha (TNF-α) concentration group in W3 ((**G**); Appendix A).

**Figure 5 animals-14-02533-f005:**
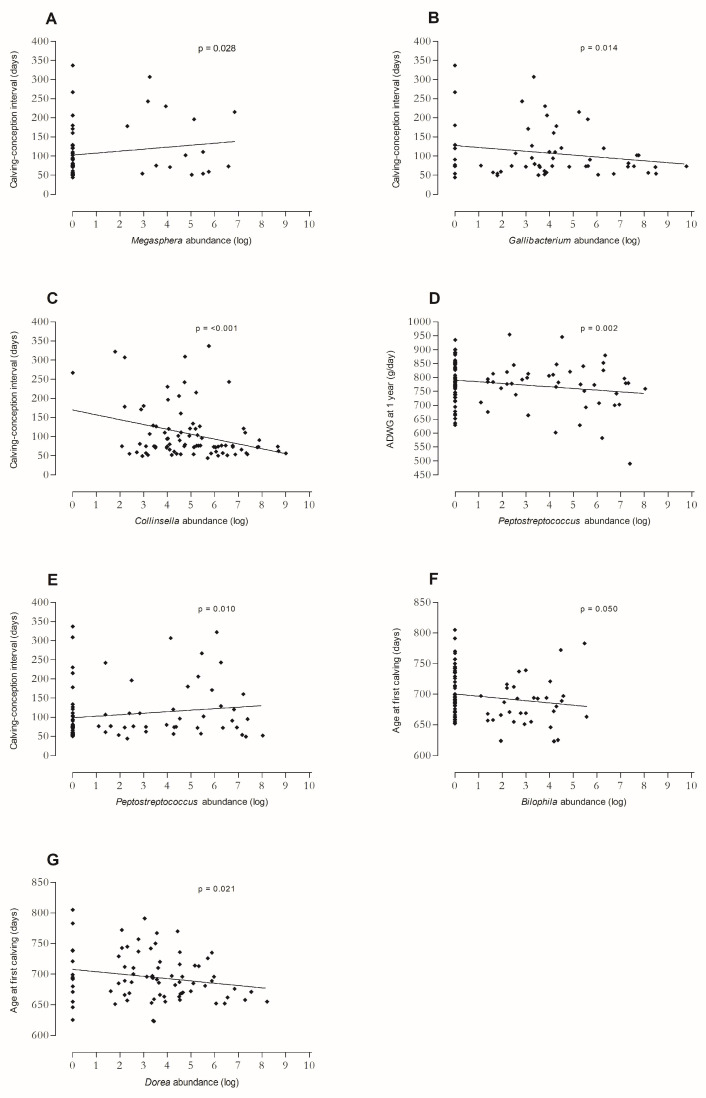
Associations between logarithmically transformed genera abundances in faecal microbiota of newborn dairy calves and performance outcomes. All associations presented in this figure were statistically significant (*p* ≤ 0.05), exact *p*-values can be found in the main text. Univariate linear regression line was included to indicate trend. Model results of multivariate regression can be found in the Appendix A. *Megasphaera* abundance and calving–conception interval in W1 (ages 1–7 days), negative binomial model (*n* = 58; (**A**); Appendix A). In the same model: *Gallibacterium* abundance and calving–conception interval in W1 ((**B**); Appendix A). *Collinsella* abundance and calving–conception interval in W2 (ages 8–14 days), negative binomial model (*n* = 84; (**C**); Appendix A). *Peptostreptococcus* abundance and average daily weight gain (ADWG) in W2, linear regression model (*n* = 95; (**D**); Appendix A). *Peptostreptococcus* abundance and calving–conception interval in W2, negative binomial model (*n* = 84; (**E**); Appendix A). *Bilophila* abundance and age at first calving (AFC) in W3 (ages 15–21 days), linear regression model (*n* = 79; (**F**); Appendix A). In the same model: *Dorea* abundance and AFC in W3 ((**G**); Appendix A).

**Table 1 animals-14-02533-t001:** Number of faecal and serum samples available for ages 1–7 days (W1), ages 8–14 days (W2), and ages 15–21 days (W3) for each outcome from heifers.

Week Groups	12 Month ADWG	AFC	305-Day-Milk Yield	1st Lactation Calving–Conception Interval	Number of Reproductive Issues ^1^
W1	67	64	64	64	67
W2	95	91	86	82	95
W3	83	79	73	72	83

ADWG—average daily weight gain; AFC—age at first calving. ^1^ Reproductive issues: records of abortion, retained placenta, endometritis.

**Table 2 animals-14-02533-t002:** Inflammatory markers in groups low, moderate, and high serum concentration for each week group of heifer calves (W1: ages 1–7 days; W2: ages 7–14 days; W3: ages 15–21 days), n_s_ indicates the number of samples.

		W1 (n_s_ = 67)	W2 (n_s_ = 95)	W3 (n_s_ = 83)
SAA (mg/L)	Low	55.2–112.6 (22)	34.0–101.2 (31)	13.2–59.1 (28)
	Moderate	113.0–159.1 (22)	102.1–154.3 (32)	60.4–103.1 (28)
	High	164.3–347.7 (23)	156.5–487.9 (32)	103.8–316.7 (27)
Hp (mg/L)	Low	97–168 (23)	119–196 (31)	85–164 (28)
	Moderate	170–260 (22)	199–716 (32)	165–400 (28)
	High	262–1899 (22)	732–2830 (32)	403–2416 (27)
IL-6 (ng/L)	Low	2.5–6.4 (23)	2.5–4.2 (32)	2.5–6.3 (29)
	Moderate	6.6–14.6 (22)	4.3–9.4 (32)	6.6–10.8 (27)
	High	16.2–82.2 (22)	9.7–130.7 (31)	10.9–43.6 (27)
TNF-α (ng/L)	Low	70–310 (22)	50–210 (32)	50–110 (29)
	Moderate	330–660 (22)	220–380 (31)	120–220 (26)
	High	670–2800 (23)	390–4200 (32)	230–910 (28)

SAA—serum amyloid A; Hp—haptoglobin; IL-6—interleukin-6; TNF-α—tumour necrosis factor-alpha.

## Data Availability

The data that support the findings of this study are available from the corresponding author upon reasonable request.

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
