# Peer review of "Associations of Neonatal Dairy Calf Faecal Microbiota with Inflammatory Markers and Future Performance"

_animals, 2024, doi:10.3390/ani14172533_

Round 1

Reviewer 1 Report

Comments and Suggestions for Authors

Dear authors,

thank you for the opportunity to read your paper about gut microbiota in calves fort he first three weaks and the associated long time affects of performance. It was a pleasure form e to read it and with some comments below it was easy to follow your way of thinking and working with that topic.

Thank you.

Line 136 ff: In summary 4-6 L of milk/milk replacer isn´t that much a calf needs to grow up. Do you think this could might be an object tot he future performace as we know that calves, that get feed with 12 L milk per day or ad libitum will show excellent performance as first lactating cows.

Table 1. Is it correct that you documented more fecal samples in week 3 than in week 1 (documented in the line: 12 month ADWG) As this is correct, it is not easy to understand as you say in the line before, that some calves died.

Line 188: You said that calves were not available – where do they have been?! It is not easy to understand. Please describe it more clearly.

Table 2: Are these results (low, moderate, high) belong alwayse to the same calves through out the three weeks. Is the same calf always high in SSA in week 1,2 and 3 or ist he category changing? If yes, pleace notice that fact in the descrition of the results.

Discussion: In general you mentioned a lot of factors affecting the microbiota like Cryptosporidiosis and animal individual things. But I miss a brief discussion oft he amounts of milk and especially milk repacer that might play a role for colonization oft he microbiota as well. Is it possible to add some sentences about the effects of that confounders. Did you use them in your model?

Reviewer 2 Report

Comments and Suggestions for Authors

The aim of the paper was to explore how the faecal microbiota of new-born dairy calves were related with inflammatory markers during the first three weeks of life, and if the abundance of specific genera was associated with first-lactation performance. They found specific faecal microbiota abundance related to inflammation from the first week of life up to the third week of life. They also could find associations of bacteria genera with performance, such as calving-conception interval, average daily weight gain and age at first calving. These relationships may help in the design of prebiotic or probiotic on farm administration programs to improve performance. 

General concept comments 

Areas of weakness: the time period of the study. Although the first few weeks are crucial in terms of nutrition and health for growth and development of the calves, it would also be very interesting to extend the monitoring up to 6-7 weeks of age or until weaning. This extension is important because the microbiota may be still developing during this period and the calf immune system may be still adapting. 

They tested the hypothesis through association and not cause-effect relationships. 

Regarding methods, as this study was conducted in a commercial farm the results and conclusions are limited to that farm. Although it could lead to other studies, this one is limited in that sense. On the other hand, as it was an observational study many variables, as stated in the manuscript, were not controlled or at least known. 

Specific comments

Introduction

Line 79. Is there any data about cytokines and APPs in serum of calves beyond week 3? If si, please address it.

Material and methods

Line 141. Please specify quantity of milk replacer offered. 

Discussion

Line 372. Wha is the evidence to state that faecal microbiota alpha diversity stablished within the 3rd week? Ver 26, 46 48

Line 439. Any idea to support the differences in results?

Reviewer 3 Report

Comments and Suggestions for Authors

In this manuscript, the authors evaluate associations between calf fecal microbial genera and measures of calf inflammatory status as well as subsequent performance.

The manuscript is fairly well-written, and I especially appreciated figure 1 as a very understandable flow chart to understand use of the sequencing data. However, I think there are some important limitations to how the manuscript is presented that need to be addressed. 

- Perhaps the most important problem with the manuscript is that - even though the very first word of the title is "associations" - the authors repeatedly make causal statements in conclusions, both in the abstract and in the discussion / conclusions.  These all need to be cleaned up - correlation dose not equal causation. The conclusion is especially egregious - lines 512 - 515 are so far from what one can conclude from a study like this that it doesn't even sound like the same research.

- The authors report data for IL-6 and TNFa with kits that are not at all validated, to this reviewer's knowledge. There are many ELISA that generate meaningless data, and without the authors sharing spike-in recovery data or at least linearity of dilution validation (or citing papers that do so), the reported values cannot be trusted. Additionally inter- and intra-assay CV should be reported. The authors say that more detailed laboratory methods are found in Peetsalu et al., but there was no more detail there.

- The statistical analysis was fairly descriptive but would have been much easier to follow if there was just an explicit model statement (either in words or in a formula). The authors describe use of exact age in days in the models, "correct treatment" or "incorrect treatment", Crypto oocysts, and of course the core variables of interest. It is not at all clear to me how or where all of these factors were used, especially when results like Fig. 2 apparently just show raw values for each calf within each group. At least, there is no mention of LS means from a more complex modeling effort.

- Why are no statistics presented for many of the figures (2,3,5)? For figure 2, the authors claim that the plot shows that calves achieved a stable microbiome by week 3, and there is absolutely no evidence for this. Not only are statistical comparisons not made, there was no sampling after week 3, so it's not even theoretically possible to show that stability was achieved then. For figure 5, please add P values (not just saying all are < 0.05) and R-squared values for each relationship. In figure 4, some of the claims of statistical significance are just impossible to believe, like medium vs .high in panel C and low vs. medium in panel D. The authors need to look at some of those again. 

Minor comments:

- The in-text references are by author last name, but the reference list is numbered, making it frustrating to find cited paper.

- Line 135: Do the authors means > 1.035? Otherwise this makes no sense to me.

- Line 51: "Immune priming" isn't really synonymous with immune maturation.

Reviewer 4 Report

Comments and Suggestions for Authors

According to the manuscript in title of “Associations of Neonatal Dairy Calf Faecal Microbiota with Inflammatory Markers and Future Performance”. These articles are remarkably well-organized, that a great deal of significant intervention of the dairy production. Because a variety of factors, including the environment, can stress the immune system, as seen by variations in the inflammatory response in neonatal dairy calves. In my opinion, several points observed are defined as follows.

-        The journal might offer a simple summary for the overview.

-        In ABSTRACT, the author could be clarified and concisely.

o   Line21-26, the significant result would be included. Please recheck the statistical evaluation.

-        Introduction part, it could be concisely and was nearly completely re-concisely explained by thoroughly with the review literature. It is possible to emphasize the importance of the faecal microbiota of newborn calves influences the content and nutritional value of milk, even if the introductory part is mostly focused on significant studies.

-        Materials and methods, it was noted that the author needed to provide additional details and do more of this clearly.

o   Line 128, why did the author carryout of research work during 2015? Why is the number (144 vs.143) different when the same procedure was used in the previous research that was cited in 10? There is a possibility of including the duration of the research.

o   As the author notes, 144 newborn calves represented the total number of animals in this study. Why does table 1's available sample count not equal 144? Every week, all of the animals were resampled?

o   It may be seen in the figure 1 illustration that there may be newborn calves instead of heifers.

o   The experimental design and hypothesis testing you decided on, which increased response variability and generalizability, were not adequately explained or justified by the author. It can be because the criteria above the data sorting were not properly validated. A unique variation data set is required for statistical testing. The author has limited the time period to January through March of 2015; therefore it is not enough time to monitor every alteration as in phases one and two.

-        Result and discussion part, it is observed that some points need to be revised.

o   Subsections that are split into the result part and the discussion section need to be included.

-        Please double-check the references in both the intext citation and the list of references, as instructed in the journal guide. For example, in line 210 and 249 as Peetsalu et al. (2022) could be use as Peetsalu et al. [10].

Reviewer 5 Report

Comments and Suggestions for Authors

Dear authors

Greetings

I would like to present the following points to improve your article:

1. Change the title: Comparison of Neonatal Dairy Calf Faecal Microbiota with In-2 flammatory Markers, and Future Performance

2. Abstract (267 words) (1th subparagraph) The authors can improve it to be clear that the development of the immune system and of the adaptive mechanisms that allow a safer transition to the extrauterine environment and HOW they would reflect in fluctuations of inflammatory markers (a flare up or a poor response to a treatment??? example). (2th subparagraph) The authors had evaluated the  the reliability circulating concentration of the inflammaging markers??? like the levels of pro-inflammatory markers???? example). I suggest to make a short, objective and clearer abstract.

3. Keywords: Dairy cow performance is a multifaceted topic that involves various factors such as milk production, health, nutrition, and overall management practices... but we just see at the end of the introduction it is about to determine if genera showing such associations may have associations with production performance and health during the first lactation. The "during the first lactation" it is at abstract, ok... but it is not at keywords, it is not at discussion, it is not at conclusions.... just future performance (it is so wide!)

Also, I suggest to organise the Keywords: neonatal dairy calf; average daily weight 30 gain; dairy cow performance; Neonatal Dairy Calf Faecal Microbiota; acute-phase proteins (APPs); 16S rRNA gene amplicon sequencing; Laboratory and Statistical Analysis? type/test?? (Elisa, immunofluorescence, Illumina MiSeq sequencing platform, example)

4. Discussion:

The gut microbiota of calves plays a crucial role in their overall health, growth, and performance. Here’s a brief overview of how these factors are interconnected: Gut Microbiota: The gut microbiome of calves begins to develop rapidly after birth and continues to evolve, influenced by diet, age, and other factors. A healthy and diverse gut microbiota is essential for preventing gastrointestinal disorders like diarrhea. Inflammation Markers: Inflammation in calves can be indicated by various biomarkers. Disruptions in the gut microbiota can lead to increased inflammation, which negatively impacts the health and growth of calves. Performance: The performance of calves, including growth rates and overall health, is closely linked to the stability and diversity of their gut microbiota. A balanced microbiome supports better nutrient absorption and immune function1. Resistance: A healthy gut microbiota helps in building resistance against pathogens. Probiotics and other dietary interventions can enhance the gut microbiota, thereby improving the calf’s resistance to diseases. Maintaining a balanced gut microbiota is key to ensuring the optimal health and performance of calves.

#Limitations of this study include the incomplete records by farm veterinarians regarding diseases and their treatments of the animals regarding possible infections or othercauses that may have activated the calves’ APR but were not noticed. >>>>> Conclusions!!!! Conclusions and Recommendations, Limitations and Suggestions for future work. 

5. References: Most of the reference articles have much more than 5 years of publication, there are few recent articles...

Suggested citation:

1. Loch, Marina and Dorbek-Kolin, Elisabeth and Husso, Aleksi and Pessa-Morikawa, Tiina and Niine, Tarmo and Kaart, Tanel and Mõtus, Kerli and Niku, Mikael and Orro, Toomas, Associations of Neonatal Dairy Calf Faecal Microbiota with Inflammatory Markers and Future Performance. Available at SSRN: https://ssrn.com/abstract=4665751 or http://dx.doi.org/10.2139/ssrn.4665751 (?????)

2. Zhuang, Y., Liu, S., Gao, D. et al. The Bifidobacterium-dominated fecal microbiome in dairy calves shapes the characteristic growth phenotype of host. npj Biofilms Microbiomes 10, 59 (2024). https://doi.org/10.1038/s41522-024-00534-4

3. Du, Y., Gao, Y., Hu, M. et al. Colonization and development of the gut microbiome in calves. J Animal Sci Biotechnol 14, 46 (2023). https://doi.org/10.1186/s40104-023-00856-x

I hope my suggestions will help you to improve your work and make it UNIQUE

Kind regards

Comments on the Quality of English Language

Minor editing of English language required.

Round 2

Reviewer 3 Report

Comments and Suggestions for Authors

I appreciate the additional explanations. Some of the confusion on this paper stems from the presentation of very superficial graphs where the real results are in the supplement, which I am not a fan of, but I suppose is the authors' prerogative as long as the details are somewhere. 

I still have 2 major concerns.

- A couple of the false causality statements were adjusted, but the most important one in the conclusions was not. Conclusions of a study should derive from the study itself, not the literature, and should not be as speculative as some of the discussion can be. The sentence below is clearly not information acquired from the results of this study: "Specific bacterial genera (such as Collinsella and Gallibacterium) downregulate inflammation and increase future performance, while others, such as Peptostreptococcus, promote inflammation and decrease future performance."

- Although Fig 4 and 5 are meant to be simple presentations of complex modeling efforts, this is not clear enough. The statement added to the Fig. 5 legend is needed in FIg. 4 as well. Furthermore, it would be much easier for readers to track if the description for each panel referred to the specific supplemental table that supports the particular graph.

Author Response

Dear reviewer,

thank you for your feedback.

Comment 1:  A couple of the false causality statements were adjusted, but the most important one in the conclusions was not. Conclusions of a study should derive from the study itself, not the literature, and should not be as speculative as some of the discussion can be. The sentence below is clearly not information acquired from the results of this study: "Specific bacterial genera (such as Collinsella and Gallibacterium) downregulate inflammation and increase future performance, while others, such as Peptostreptococcus, promote inflammation and decrease future performance."

Reply 1: The conclusions have been adjusted to the following: 

"The observations in this study suggest that early-life faecal microbiota can influence dairy cow performance via immunomodulation. It can be assumed that specific bacterial genera (such as Collinsella and Gallibacterium) are able to downregulate inflammation and increase future performance, while others, such as Peptostreptococcus, can promote inflammation and decrease future performance. As the associations found in this study change over the first three weeks of life, the timing of colonization with specific bacteria seems to play a critical role."

Comment 2: Although Fig 4 and 5 are meant to be simple presentations of complex modeling efforts, this is not clear enough. The statement added to the Fig. 5 legend is needed in FIg. 4 as well. Furthermore, it would be much easier for readers to track if the description for each panel referred to the specific supplemental table that supports the particular graph.

Reply 2: We have added the information to the figure legends as suggested.

Figure 4. Logarithmically transformed abundances of faecal microbiota genera by inflammatory marker concentration category (median and interquartile range). Italic numbers above the X-axis depict sample number per concentration category. Bonferroni-corrected p-values are given for statistically significant differences in genus abundance between concentration groups. Model results of multivariate regression can be found in the supplemental tables (ST). Megasphaera abundance in W1 (ages 1-7 days) by interleukin-6 (IL-6) concentration group (A; ST 3). Gallibacterium abundance in W1 by IL-6 concentration category (B; ST 4). Collinsella abundance in W2 (ages 8-14 days) by serum-amyloid A (SAA) (C; ST 7) and haptoglobin (Hp) (D; ST 8) concentration category. Peptostreptococcus abundance by Hp concentration category in W2 (E; ST 11). Dorea abundance by Hp concentration group in W3 (ages 15-21) (F; ST 16). Bilophila abundance by tumour necrosis factor alpha (TNF-α) concentration group in W3 (G; ST 19).

Figure 5. Associations between logarithmically transformed genera abundances in faecal microbiota of newborn dairy calves and performance outcomes. All associations presented in this figure were statistically significant (p ≤ 0.05), exact p-values can be found in the main text. Univariate linear regression line has been included to indicate trend. Model results of multivariate regression can be found in the supplemental tables (ST). Megasphaera abundance and calving-conception interval in W1 (ages 1-7 days), negative binomial model (n = 58; A; ST 6). In the same model: Gallibacterium abundance and calving-conception interval in W1 (B; ST 6). Collinsella abundance and calving-conception interval in W2 (ages 8-14 days), negative binomial model (n = 84; C; ST 14). Peptostreptococcus abundance and average daily weight gain (ADWG) in W2, linear regression model (n = 95; D; ST 15). Peptostreptococcus abundance and calving-conception interval in W2, negative binomial model (n = 84; E; ST 14). Bilophila abundance and age at first calving (AFC) in W3 (ages 15-21 days), linear regression model (n = 79; F; ST 20). In the same model: Dorea abundance and AFC in W3 (G; ST 20).

We understand that we cannot claim to know about cause and effect in or study, but would still like to emphasize the potential impact of our findings in the conclusions. We hope that the revised version is an acceptable compromise.

Kind regards,

Marina Loch

on behalf of all authors